# T Cells in Tick-Borne Flavivirus Encephalitis: A Review of Current Paradigms in Protection and Disease Pathology

**DOI:** 10.3390/v15040958

**Published:** 2023-04-13

**Authors:** E. Taylor Stone, Amelia K. Pinto

**Affiliations:** Department of Molecular Microbiology and Immunology, Saint Louis University School of Medicine, Saint Louis, MO 63103, USA

**Keywords:** T cells, tick-borne flaviviruses, Powassan, tick-borne encephalitis virus

## Abstract

The family *Flaviviridae* is comprised of a diverse group of arthropod-borne viruses that are the etiological agents of globally relevant diseases in humans. Among these, infection with several of these flaviviruses—including West Nile virus (WNV), Zika virus (ZIKV), Japanese encephalitis virus (JEV), tick-borne encephalitis virus (TBEV), and Powassan virus (POWV)—can result in neuroinvasive disease presenting as meningitis or encephalitis. Factors contributing to the development and resolution of tick-borne flavivirus (TBEV, POWV) infection and neuropathology remain unclear, though many recently undertaken studies have described the virus–host interactions underlying encephalitic disease. With access to neural tissues despite the selectively permeable blood–brain barrier, T cells have emerged as one notable contributor to neuroinflammation. The goal of this review is to summarize the recent advances in tick-borne flavivirus immunology—particularly with respect to T cells—as it pertains to the development of encephalitis. We found that although T cell responses are rarely evaluated in a clinical setting, they are integral in conjunction with antibody responses to restricting the entry of TBFV into the CNS. The extent and means by which they can drive immune pathology, however, merits further study. Understanding the role of the T cell compartment in tick-borne flavivirus encephalitis is instrumental for improving vaccine safety and efficacy, and has implications for treatments and interventions for human disease.

## 1. Global Public Health Burden of Tick-Borne Flavivirus Encephalitis

Encephalitic flaviviruses have been isolated on almost every continent [1]. Although the extent to which these flaviviruses contribute significantly to the global health burden is variable, they nonetheless pose a significant public health threat. Up to 400 million people are infected with flaviviruses annually; of these, tick-borne flaviviruses in particular have the potential to cause severe neuroinvasive disease [2]. Many of these are infections that are predicted to increase as the climate continues to warm and the range of arbovirus vectors such as mosquitoes [3,4] and ticks [5,6,7] continues to expand. Already substantial, the risk of encephalitic flavivirus infection therefore has the potential to further increase as the global climate continues to change. Many mosquito-borne flaviviruses have been studied in detail. Among encephalitic flaviviruses, both the West Nile virus (WNV) and Zika virus (ZIKV) have been extensively studied, especially following their emergence in the Western Hemisphere in 1999 and 2016, respectively. Less well-studied are the tick-borne flaviviruses, such as tick-borne encephalitis virus (TBEV) and Powassan virus (POWV). This review will focus on summarizing tick-borne flaviviruses (TBFVs) and what is currently known about their propensity to cause encephalitis, with an emphasis on the activation, recruitment, and functionality of the T cell response during TBFV infection.

### 1.1. Tick-Borne Encephalitis Virus (TBEV)

TBEV is transmitted to humans from the bite of *Ixodes* ticks of numerous species and is endemic in Europe and Asia. Though possibly described as early as the 1700s [8], the first widely accepted description was given in the 1930s by Soviet Union scientists [9]. There have been reported incidents of humans becoming infected not only from infected ticks, but also via the consumption of raw milk from TBEV-infected animals (reviewed in [10]), or due to the transplantation of solid organs from infected persons [11]. Humans are considered a dead-end host for TBEV, but several woodland mammals have been suggested as natural reservoirs for TBEV. Evidence strongly implicating a single animal in the transmission cycle, though, has been limited (reviewed in [12,13,14]). It should also be noted that vertical transmission within tick populations may be important in perpetuating the TBEV lifecycle [15,16]. though the contribution is likely small relative to other routes or in mammalian hosts [17]. Co-feeding—or nonviremic transmission (NVT) facilitated by tick saliva—is thought to be particularly important in the maintenance of the TBEV life cycle, allowing for the minimal reliance on replication in a mammalian host [18,19]. In TBEV endemic regions, the case numbers can vary from season to season, but cumulatively have ranged from 1,500–4000 annually between 2000–2019 [20]. TBEV exists in three to seven recognized subtypes depending on classification schema, with the European (TBEV-Eu), Siberian (TBEV-Sib), Far-Eastern (TBEV-FE), Baikalian (TBEV-Blk), and Himalayan lineages (TBEV-Him), being attributed to distinct disease outcomes [20,21,22,23].

TBEV disease in humans can range from asymptomatic to severe, and mortality rates are dependent upon the infecting subtype. Although TBEV infection occurs within a large geographic area, TBEV incidence tends to occur in focal hot spots across a given region [24]. Within these hot spots, it has been demonstrated that there is a potential for asymptomatic cases [25], making TBEV incidence likely an underestimation [25,26]. Moreover, disease reporting can vary across the many countries in which TBEV is endemic. For TBEV-Eu, the reported mortality rates range from 2–4% [27], while for TBEV-FE they have been reported to be as high as 30% [28]. Though there is less information available for the other subtypes, TBEV-Sib has been reported to cause only a mild, nonparalytic febrile illness [29]. Although several efficacious vaccines exist for TBEV, the vaccine coverage is not homogenous, and can be low in some TBEV endemic areas [25,30]. There are no approved antivirals for human use [31], although some antivirals have been shown to be efficacious in murine models (reviewed in [30]).

In humans, TBEV infection can present as biphasic, where the first viremic phase often causes mild febrile illness, including headache, fever, nausea, and vomiting. For many patients, the illness remains monophasic and the infection is resolved in the absence of neurological symptoms or involvement. Following the first viremic phase, though, roughly one-third of patients will experience a second phase characterized by complications due to neurological involvement [32]. Neurological illness is further subdivided into TBEV with meningitis, TBEV without paralysis, and TBEV with paralysis. In general, TBEV illness is more severe in elderly populations [32,33].

### 1.2. Powassan Virus (POWV)

POWV is transmitted to humans from the bite of *Ixodes cookei* or *Ixodes scapularis* ticks and is endemic in North America and the Russian Far East, where it can also be transmitted by *Dermacentor* ticks [34,35,36]. It was originally described in 1958 after causing a fatal case of encephalitis in a young boy in Powassan, Ontario [37].

POWV exists in two major lineages, known as lineage I (POWV-LB) and lineage II (POWV-Spooner). They are thought to be maintained in separate tick and mammalian host species (summarized in [38]). In general, POWV-LB is thought to be maintained by *Ixodes cookei* ticks and woodland mammals such as woodchucks. This lineage more rarely infects humans, as *Ix. cookei* ticks are nidicolous and therefore considered to be less likely to take bloodmeals on humans. POWV-Spooner—sometimes referred to as deer tick virus or DTV—is maintained by *Ixodes scapularis* ticks, which are more promiscuous feeders. For this reason, a majority of human POWV disease in the United States can be attributed to POWV-Spooner.

The incidence of POWV in humans is rare, with only 6–39 cases reported annually in North America between 2011 and 2020 [39]. Notably, this number has begun to increase, with just six cases in 2015 rising to 20 cases in 2020 [40]. Furthermore, the range and overwintering ability of POWV vectors are forecasted to increase as the climate warms [5,6,7], making more cases likely. POWV disease ranges in severity from asymptomatic to severe cases. The contribution of these asymptomatic cases is not well understood, and the case numbers likely represent an underestimate of POWV infection [41]. POWV cases have a relatively high mortality rate of 10–15%, with some 50% of recovered patients suffering from some form of long-term neurological sequelae [42]. Moreover, there are currently no approved antivirals or therapeutics for POWV disease in humans.

## 2. Flavivirus Structure and Replication

Flaviviruses such as TBEV and POWV have a common structure that includes an ~11 kb positive sense RNA genome. This genome encodes three structural proteins: the capsid, pre-membrane, and envelope (C, prM, E), and seven non-structural proteins (NS1, NS2A, NS2B, NS3, NS4A, NS4B, and NS5). In addition, the genome also encodes 5′ and 3′ untranslated regions (UTRs). This positive sense RNA is translated as one long polyprotein, which is then cleaved by both viral and host proteases to form the ten subsequent proteins required for the viral life cycle. Entry is mediated by binding with one of multiple possible host receptors, followed by receptor-mediated endocytosis. In the acidic endosome, the virion is then uncoated and the release of genetic material into the cytoplasm occurs following endosomal escape. Following translation, the synthesis of new genomic RNA, and polyprotein processing, progeny virions can then assemble in the endoplasmic reticulum and mature through the Golgi until their eventual fusion and release. Importantly, the maturation state of the progeny virions can impact not only the infectivity, but also the host antibody responses due to confirmational changes that dictate the accessibility of antigenic sites on the envelope protein, the main antigenic determinant for flaviviruses [43]. The non-structural proteins, in addition to being indispensable for viral replication, are known to be targets for both the T cell [44] and antibody responses [45,46,47].

## 3. Meningitis, Encephalitis, Myelitis, Encephalomyelitis, and Meningoencephalitis

During flavivirus infection of the central nervous system (CNS), inflammatory disease is often present and pathologically distinguished by the affected regions of the CNS. Inflammation of the membrane surrounding the brain and spinal cord—termed the meninges—is referred to as meningitis [48]. This is distinct from inflammation of the brain parenchyma, which is referred to as encephalitis, or inflammation of the spinal cord, which is referred to as myelitis [48]. In cases where both the brain and spinal cord are inflamed, the term encephalomyelitis is used [48]. Occasionally, the term encephalitis will be used to refer collectively to the inflammation of both the brain and spinal cord. Therefore, the term meningoencephalitis has also been used to indicate inflammation of the meninges, brain parenchyma, and spinal cord. Though these terms distinguish the anatomic sites within the CNS, they are not indicative of the severity of disease. In general, meningitis is considered to be milder than encephalitis or myelitis, wherein infection of the brain parenchyma is considered to be more severe. Both TBEV and POWV have the capacity to cause meningitis, encephalitis, and myelitis in humans, and severe infection of the brain parenchyma is possible.

## 4. T Cell Responses Early in TBFV Infection

### 4.1. TBEV

Among flaviviruses, TBEV has some of the most extensively described models of immune pathology. Furthermore, in addition to being re-capitulated well in disease models, evidence of immune pathology has been noted in human cases of severe encephalitis. In addition to describing the CD8+ T cell response to TBEV, this section will highlight areas where CD8+ T cells have been implicated in immune pathology.

In humans, acute symptomatic illness due to TBEV is considered to be biphasic (Figure 1). The first phase (acute) consists of general viral syndromic illness, including fever, headache, dizziness, and malaise, and occurs in the first 7–10 days. A second phase occurs 15–28 days after infection and is characterized by fever and CNS inflammation, typically in the form of meningitis, encephalitis, or meningoencephalitis with myelitis. Because TBEV is not typically clinically diagnosed until the second phase of illness, the majority of the studies typically address T cell responses detected in the second phase of TBEV illness. Studies using an in vitro culture of primary human neurons show a high expression of chemokines important in the recruitment of CD8+ T cells into the CNS, CXCL10, CXCL11, and CCR5 as early as 24 h post-infection with the TBEV-Hypr strain [49]. It is important to note, however, that other innate immune cell types can also be important producers of these chemokines following the detection of Pathogen Associated Molecular Patterns (PAMPS) by Pattern Recognition Receptors (PRRs) [30,50,51].

The most comprehensive analysis of the CD8+ T cell response during TBEV infection was a longitudinal study of twenty clinically diagnosed TBEV patients, wherein samples were captured on days 0, 7, 21, and 90 following hospital admission [52]. All patients were symptomatic, and therefore considered to be in the second phase of the biphasic TBEV illness. Studies of the T cell responses in this second phase of illness identified TBEV-specific CD8+ T cells in the peripheral blood mononuclear cells (PBMCs) of humans infected with the European TBEV subtype [52]. In addition to identifying an HLA-A2-restricted CD8+ T cell epitope, the authors described that most responding CD8+ T cells were considered T effector cells (CD45RA-, CCR7-). Many (>50%) of the cells were monofunctional and contained strong T-bet and Eomes transcriptional signatures, while also expressing low levels of Bcl-2 and high levels of Ki67+. Many of these cells were also PD-1+. Of these, Eomes^+^ CD8+ T cells were also the more functional, producing higher levels of effector molecules granzyme B and perforin than Eomes^-^CD8+ T cells. In terms of the kinetics of this response, the production of cytokines IFN-γ and TNF-α peaked 21 days following hospitalization.

In a similar follow-up study conducted by the same group, HLA-A2 and HLA-B7 epitopes were identified, and tetramers were developed to examine the kinetics of the TBEV T cell response [44]. Consistent with previous studies, on admission, no antigen-specific tetramer-positive cells were observed (day 0). TBEV-specific responses peaked between days 7 and 21 post-hospitalization, with up to 1.16% of CD8+ cells being tetramer-positive for a specific HLA-B7 (B7-6) epitope. Interestingly, one HLA-A2+ patient mounted a robust 2–3% response to a specific HLA-A2 epitope (A2-19), which was not detected in the other four HLA-A2+ patients. Tetramer-positive CD8+ T cells had increased CD27 (TNF receptor superfamily protein) expression relative to tetramer-negative CD8+ T cells. Tetramer-positive CD8+ T cells also did not express high levels of CD103, which has been used to characterize populations of brain-tissue-resident memory T_RM_ cells [53], CCR5, or CCR6. The α4 and β1 integrins were highly expressed on all tetramer-positive CD8+ T cells but were also very high on tetramer-negative CD8+ T cells and T cells from healthy control donors. CXCR3 expression was notably different among different donors. Meanwhile, whereas some patients had the CXCR3 expression peak on day seven and decrease by day 21, others saw the peak expression on day 21. As relatively little information was given on the clinical course of the disease among donors, it is difficult to deduce how changes in the kinetics of the CD8+ T cell response may impact the disease progression. Nevertheless, these studies serve as the most in-depth characterization of the CD8+ T cell compartment over time in a cohort of human TBEV patients. They suggest that the CD8+ T cell response is primed and active during the biphasic portion of the disease, as well as the resolution of the disease.

In studies of TBEV infections in *Rhesus macaques*, inoculation by both subcutaneous and intracranial routes with three different isolates and two different TBEV mutants resulted in variable outcomes in the macaques [54,55,56]. The outcomes ranged from asymptomatic meningitis to meningoencephalitis with myelitis and persistent infection. Notably, the antibody responses assessed in these experimental infections were detectable shortly after the challenge, indicating a productive infection [57]. In one study of a naturally infected monkey (*Macaca sylvanus*) the authors demonstrated successful isolation of TBEV from the brain [58]. The infection of Purkinje cells was also detected, a cell type that is susceptible to TBEV infection in humans.

Due to their tractability, murine models have been helpful in assessing pathology, immune correlates of protection [59,60,61,62,63], as well as in vivo antiviral [64] and vaccine efficacy [60,62,63]. Acute TBEV infection in mice is characterized by initial rounds of replication in the skin and Langerhans cells followed by dissemination into secondary lymphoid tissues, as observed in outbred mice, which can support TBEV replication [18,19]. In laboratory mouse strains, C57BL6 and BALB/c mice develop a similar disease progression as in humans, with hallmarks of biphasic disease, meningitis, and encephalitis, and typical resolution that confers protective immunity [60,65]. Of note is that BALB/c mice appear to have slightly better morbidity and mortality outcomes following TBEV infection, coinciding with higher levels of IFN-γ production relative to TNF-α production, compared to C57BL/6 mice [65]. The same pattern held for chemokines such as RANTES, MIP-1α, and MIP-1β.

Regarding the T cell compartment, some studies have examined TCR usage in C57BL/6 mice predicted to survive or succumb to TBEV infection. In this murine model of TBEV-FE infection (Oshima strain), the analysis of TCR usage in the brain at 13 DPI between surviving and dying mice reported a high frequency of the *VA15-1/AJ12* and *VB8-2/BJ1.1* gene usage in dying mice [61]. Conversely, *VA8-1/AJ15* or *VA8-1/AJ23* gene usage was more common in surviving mice. Despite these significant differences, there were no discernable differences in surviving vs. dying mice with respect to the TBEV genome copy number or differential expression of CD8+ T cell activation markers. Although the article reported statistically distinct TCR gene usage, attributing a particular clonotype to either viral control or host pathology will require further studies.

In mice it has been shown that CD8+ T cells mediate immunopathogenesis following TBEV infection [59]. Specifically, these studies showed that the adoptive transfer of naïve or TBEV immune CD8+ T cells into SCID mice challenged with TBEV drastically reduced the survival time following the TBEV-Hypr challenge. SCID and CD8KO mice also displayed prolonged survival times relative to immune-competent BALB/c and C57BL6 mice, implying that adaptive immune deficiency—or lacking a CD8+ T cell compartment—is sufficient to delay TBEV mortality, though notably, all mice still succumb to the disease. Only in the case of the adoptive transfer of CD4+ T cells into SCID mice was partial protection achieved. This observation was in line with previous findings that the transfer of splenocytes from TBEV-infected mice shortened the incubation period and that immunosuppressed mice increased the mean survival time following TBEV challenge [66]. Interestingly, viral titers in CD8KO mice tracked closely with that of WT mice in the spleen and sera of TBEV-challenged mice, implying that CD8+ T cells were not crucial in the peripheral clearance of the virus. It is also worth noting that CD4+ Tregs have been described as having a role in mitigating immune pathology due to flavivirus CNS infection [67], but the extent to which this phenomenon could be occurring in TBFV infection requires further study.

### 4.2. POWV

Acute POWV infection is characterized by initial rounds of replication at the tick–host interface, likely in Langerhans cells and tissue-resident monocytes [68] before dissemination to secondary lymphoid tissues and eventually entrance into the CNS. For POWV, relatively few studies have examined the protective capacity of T cells, but T cell responses can be detected in murine models of infection [69] and vaccination [70]. Much like TBEV, there have been minimal studies of POWV infection in non-human primates. However, the infection of macaques with POWV has indicated that POWV can establish a lytic infection of neurons in the brain and spinal cord [71].

Using the infection of primary human tissues in vitro, it has been established that POWV can productively infect several cell types. In addition to the ability of POWV to infect hBMECs, POWV is also known to infect human primary neurons and pericytes [72]. Regarding cytokine responses in POWV-infected cells, it has recently been shown that the POWV infection of primary human neurons results in relatively low levels of cytokine production relative to WNV infection [73]. Although inflammatory cytokine levels in this in vitro system remained low from 0–72 h post POWV infection, the authors did note an induction of chemokines, which they speculated may be important for recruiting activated lymphocytes and subsequent tissue destruction. Notably, the authors observed that the POWV infection of primary human neurons resulted in relatively little apoptosis compared to WNV, which suggests that this feature may be unique to TBFVs [73].

Outbred mice challenged with POWV typically present with a very transient viremia, no overt signs of disease, and little antigen detected in the CNS [74]. Though the exact reasons for this are unknown, it has been documented that—in outbred *P. leucopus* mice—antigen peptide transporter 1 (TAP1) mRNA is highly upregulated in the brain from 1–7 DPI [75]. As TAP1 upregulation is critical for peptide loading onto class I MHC, this suggests that natural hosts surviving POWV infection are primed to mount a CD8+ T cell response in the CNS very early after infection.

In bred strains of laboratory mice including BALB/c and C57BL/6, POWV viremia is transient, typically resolving by 2–3 DPI before dissemination to lymphoid tissues and eventually the CNS [76]. In contrast to the TBEV findings, studies with POWV challenge have demonstrated that BALB/c mice have slightly worsened survival outcomes following the challenge with a similar dose of the POWV-LB lineage [74]. Regarding the T cell responses to POWV, in the case of stringent POWV-LB challenge, tetramer-positive CD8+ T cells are detectable in the peripheral blood of C57BL/6 mice by seven days post-infection [77]. Furthermore, tetramer-positive cells can also be isolated from the brains of POWV-LB- and POWV-Spooner-challenged mice at eight DPI, and have been shown to produce Granzyme B, IFN-γ, and TNF-α following stimulation with a cognate antigen. CD4+ T cells also made IFN-γ and TNF-α in response to peptide stimulation. Although it appears that an effector CD4+ and CD8+ T cell response is successfully primed and mounted, it is insufficient to control POWV-LB challenge. How this response may differ in the context of a less lethal POWV-Spooner challenge remains to be elucidated. Notably, in our hands, we have seen high proportions of CD8+ T cells infiltrating the brains of POWV-LB challenged mice, perhaps unsurprising given that we also observe higher titers at earlier time points following POWV-LB infection. Interestingly, the adoptive transfer of T cells from POWV-Spooner immune donors did not protect B6 mice. In our depletion studies, we found that the depletion of CD4+ T cells did not impact the survival during POWV-LB infection, but that depletion of CD8+ T cells resulted in slightly prolonged survival. Although all mice eventually succumbed to infection, this finding is in accordance with murine models of TBEV infection, wherein CD8+ T cells may mediate the immune pathology, and their absence during murine infection results in delayed mortality relative to wild-type mice [59]. Further studies are needed in murine models, as well as the human POWV disease setting, to better understand the role of the CD8+ T cell compartment in POWV disease.

## 5. Trafficking to and across the Blood–Brain Barrier into the CNS

The blood–brain barrier (BBB) is a selectively permeable interface that separates the immune-privileged CNS from the rest of the organism. For neuroinvasive flaviviruses, exclusion by the BBB must be overcome in order for replication to occur within the permissive cell types of the CNS. The BBB is itself comprised of three primary cell types: endothelial cells, pericytes, and astrocytes. Cytokines and chemokines produced during TBFV infection are known to act on many of the cell types present at the BBB to alter its permeability. Certain cytokines such as IL-1a [78], IL-6 [79], and TNF-α [80] are accessible to the CNS via the BBB and have been shown to act synergistically in mouse neurons to cause neuronal damage [81]. Furthermore, it is well-established that either IL-6 or TNF-α administered to human brain microvascular endothelial cells in vitro causes increased permeability [79,82]. Other cytokines that are not able to cross the BBB (i.e., IL-2, [83]), however, must be produced by cells already localized to the CNS or produced by cells recruited there.

It is important to note that the crossing of the BBB and the subsequent CNS infection are not due to failure of the host to mount an antibody response. Several studies of TBEV have demonstrated robust antibody responses throughout and following infection, and TBFV-specific IgM is a commonly used diagnostic criteria [41,84,85]. Due to the selective permeability of the BBB, this is typically only observed in the cerebrospinal fluid (CSF) when the BBB is compromised.

The BBB is organized to prevent neurotoxins and pathogens from gaining access to the CNS. Pathogens are nevertheless able to bypass the BBB through a variety of mechanisms. Among these are paracellular and transcellular entry, as well as ‘Trojan horse’ entry [86], retrograde axonal transport [87], and via infection of the olfactory bulb [88,89]. Flaviviruses are thought to primarily enter the CNS via a hematogenous route of spread by utilizing paracellular and transcellular entry to gain access to the CNS, although more studies in this area are needed. Figure 2 shows the likely mechanisms by which TBFVs access the CNS tissue.

### 5.1. TBEV

The BBB is known to be disrupted during the course of human TBEV disease [90]. The exact mechanism by which TBEV infection causes BBB disruption has not been fully elucidated. It has been demonstrated in murine models of infection that BBB disruption typically follows high viral titers in the brain and is not required for TBEV to reach the CNS [65]. Figure 2 shows the approximate course of TBFV disease in humans as it relates to the BBB integrity and T cell responses. Although current evidence suggests that BBB disruption is not required for the success of WNV [91] or TBEV [65] infection of the CNS, the BBB undoubtedly represents a unique challenge for mounting an effective host immune response against TBFVs that have established CNS infection. Regarding receptor binding, literature implicating a single receptor as necessary and sufficient for TBFV infection of the CNS, is scant. However, both laminin binding proteins and integrins have been implicated as possible receptors for TBEV [92,93,94,95,96].

TBEV has been shown to establish a productive infection of cells comprising the BBB, including the human brain microvascular endothelial cells (hBMECs) [97], pericytes [98], and astrocytes [49,98,99]. Studies of gene expression in hBMECs treated with recombinant envelope protein (domain III, rDIII) from TBEV have revealed an upregulation of genes associated with virus uptake, tight junction disruption, extracellular matrix reorganization, and the activation of innate immunity [100]. Although apoptosis was also implicated in these studies, TBEV generally elicited a more modest upregulation of genes associated with cytokine/chemokine signaling, as well as apoptosis relative to a WNV comparator [100]. Other studies of the TBEV-Hypr infection of primary human neurons and astrocytes have suggested that both cell types can be an important source of pro-inflammatory cytokines, although these responses were modest in human neuronal cells compared to astrocytes [101]. These findings suggest that the induction of inflammation and apoptosis by TBFVs may differ mechanistically from that of mosquito-borne flaviviruses such as WNV.

Notably, CD8+ T cells are not required for BBB permeability during TBEV infection. Studies of murine models of TBEV infection have demonstrated that BBB disruption occurs independent of CD8+ T cells, as the infection of CD8 knockout mice did not impact the BBB permeability as measured by sodium fluorescein [65]. Interestingly, CD8a KO mice lacking CD8+ T cells still presented with high levels of IFN-γ and TNF-α, as well as pro-inflammatory cytokines in the brain, implying that other innate immune cells and supportive cells of the CNS can contribute to the inflammatory cytokine milieu during TBEV infection [65]. This finding was in accordance with the observation that BBB disruption typically occurs at a time when CNS viremia is already high, implying that the BBB permeability was neither reliant on CD8+ T cells, nor was TBEV neuroinvasion reliant upon the BBB permeability. However, CD8+ T cells have been well-documented to be necessary for the disruption of the BBB during other viral infections [102], implying that the mechanisms at work in TBFVs are likely distinct.

### 5.2. POWV

There are several proposed mechanisms of entry for POWV into the CNS, including (1) crossing the BBB following transient viremia, (2) a ‘Trojan horse’ mechanism of entry requiring replication in CNS-infiltrating leukocytes, and (3) retrograde entry following the infection of peripheral nerves. Which of these is the most common means of entry to the CNS for POWV is an area of active debate, though some studies have indicated that this is dependent upon the dose and factors related to the immune status of the host, or host–vector interactions [103,104].

Disruption of the BBB has not been clearly demonstrated in human POWV infection, though high POWV-IgM levels in the cerebrospinal fluid are a common diagnostic tool. Moreover, is it not known whether BBB disruption is required for POWV to reach the CNS. POWV has, however, recently been shown to infect human brain microvascular endothelial cells and pericytes [72]. These cell types—located at the BBB—showing susceptibility and permissivity to infection, represent one possible mechanism by which POWV enters the CNS and establishes infection. Notably, imaging studies of the brain in murine POWV infections did not detect significant levels of POWV in the olfactory bulb, suggesting that this route may not be as important in POWV neuroinvasion [104]. However, more mechanistic studies in this area are needed.

## 6. The Central Nervous System (CNS) and Immune-Privilege

The brain has long been considered an immune-privileged site owing to the selective permeability of the BBB to immune cells, minimal expression of major histocompatibility complex (MHC) in CNS tissues, and limited understanding of the CNS lymphatics. It was long thought that the CNS contained no lymphatic structures. Studies demonstrating failures to reject allogenic tissue grafts in the CNS reinforced the notion that the CNS was isolated from the immune system [105]. However, it has more recently been appreciated that lymphatic structures can be observed in the CNS [106,107], spurring a new era of inquiry into the system of lymphatic drainage in the CNS and its implications for the treatment of CNS infection and injury.

The recognition of lymphatic structures in the CNS therefore provides two mechanisms by which CD8+ T cells can interact with cognate antigens during viral infection: (1) the recognition of antigen in the periphery, followed by recruitment into the CNS after the pathogen has replicated in the CNS tissue, and (2) the recognition of antigen from the CNS via lymphatic drainage (Figure 3, reviewed in [108]). In the case of TBEV and POWV infection, the former is more likely due to primary rounds of viral replication in the skin and lymphoid tissue following the bite of an infected vector [18,19,68,76].

Resident immune cells such as microglia also have an established role in the activation of CD8+ T cells during flavivirus encephalitis [109,110]. Indeed, there has even been some suggestion that a feed-forward mechanism exists between microglia and CD8+ T cells, wherein proinflammatory cytokine production (i.e., IFN-γ) drives CXCR3 and CCR2 ligands to promote neuronal damage [110]. Moreover, during the intracranial infection of mice with an attenuated WNV strain (E218A), mice lacking microglia showed a normal recruitment of CD8+ T cells into the CNS but reduced activation of WNV-specific T cells as measured by CD69 expression. The authors attributed this reduced activation to a reduction in co-stimulatory molecule expression (i.e., CD86) by APCs in the draining lymph node [109]. Though there is likely redundancy in the mechanisms of recruitment and activation of T cells into the flavivirus-infected CNS, there is undoubtedly a high degree of cross-talk between CNS-resident immune cells and T cells that inform the effector functions utilized in attempts to control infection. Moreover, the activation of innate immunity via PAMPs and PRRs is likely to potentiate the activation and recruitment of T cells into the CNS via cytokine and chemokine production, and is overall critical for inducing an antiviral state [111,112,113].

Class I MHC is required for the presentation of viral antigen in order to activate CD8+ T cells and exert their cytolytic killing effector functions at the site of infection [114]. Requiring an MHC-I α-chain and β2 microglobulin (β2m) for stable surface MHC-I expression, the signal strength and duration of the immune synapse formation is critical for the activation of cytotoxic T cell responses, but is also critical for determining the position and likelihood of degranulation [115]. Previously, it was thought that neurons either did not express—or expressed very little—detectable MHC-I [116]. This has been posited as a mechanism to explain persistent viral infections of the brain [117] together with inefficient peptide loading on MHC-I owing to the low expression of transporters associated with antigen processing (TAP1 & TAP2, [118]) and β2m [119]. Despite a low expression of MHC-I under homeostatic conditions, viral infection—by contrast—has been shown to upregulate MHC-I expression on cell types in the CNS such as astrocytes [120], oligodendrocytes [121], brain endothelial cells [122], and neurons [117].

One prevailing theory regarding the role of T cells in the CNS is that they mainly exert antiviral functions not by cytolytic killing, but by the production of cytokines and the induction of an antiviral state (reviewed in [123]). Interestingly, whether neurons are more susceptible to Fas/FasL or perforin/granzyme-mediated killing remains controversial [124,125]. Further complicating this theory is the fact that even non-cytolytic killing means—such as the production of proinflammatory cytokines such as IFN-γ—can cause neurons to increase the MHC-I expression and increase their susceptibility to cytolytic killing [126]. Though the exact contribution of non-cytolytic killing remains difficult to quantify, many TBFVs have mechanisms to subvert the effect of antiviral cytokines, particularly interferons [51,127,128], making it more difficult to disentangle the contribution of cytolytic vs. non-cytolytic killing. Altogether, it seems that the T cell compartment must somehow restrict the reliance on cytolytic killing in the CNS, if for no other reason than to preserve the host tissue and prevent complete neuronal loss during TBFV infection.

### 6.1. TBEV

Due to the relatively high (3000 cases annually) incidence of TBEV in humans in endemic areas, much of the information regarding the TBEV-specific T cell responses in the CNS comes from work on humans. In an analysis of post-mortem brain sections of clinically diagnosed TBEV infections, a possible role for CD8+ T cells in driving immune pathology was noted [129]. Specifically, in the 28 patients with TBEV, neurons staining positively for TBEV-antigen were detected in proximity to CD8+ T cells, which the authors cited as being potentially indicative of neuronal killing by CD8+ T cells, followed by phagocytosis by HLA-DR+ macrophages or microglia. In follow-up studies, the CD8+ T cells found in close proximity to TBEV-antigen+ neurons were also found to be Granzyme B+ [130]. Based on these results, the authors attributed cytotoxic T cell infiltration and the hyperactivation of macrophages and microglia as important drivers in severe clinical cases of TBEV. Again, this raises the question of the most important mechanisms of CD8+ T cell killing during TBEV disease.

The CCR5/CCL5 signaling axis has recently been of particular interest during TBEV infection for its role in the recruitment of lymphocytes to the CNS. There has been some suggestion that in human TBEV infection, the deletion of the chemokine receptor CCR5 is associated with more severe disease [131]. This receptor is highly expressed on T cells, and serves as a receptor for CCL3, CCL4, and CCL5. Furthermore, CCR5 has been implicated in numerous neurotropic flavivirus infections (reviewed in [132]). In particular, CCL5 is highly upregulated in the CSF of TBEV patients [133], and in mice it is upregulated in an interferon-dependent (IRF3) manner [127]. However, more recently published studies found no association between the CCR5Δ32 allele and severe disease in a subset of TBEV patients [134]. Moreover, CCR5 was not found to be highly expressed on TBEV-specific tetramer-positive CD8+ T cells during TBEV infection [44]. It has also been shown that following TBEV neuroinvasion, mice produced high levels of CXCL10 [101]. Intriguingly, in mice with differential susceptibility to TBEV-Neudoerfl (TBEV-Eu), the susceptibility was inversely proportional to CXCL10 and MCP-1 expression (i.e., strains of mice producing the most CXCL10 by qPCR were more susceptible) [135]. Further studies are needed to determine the protective capacity of CCR5, CXCL10, and MCP-1 in human TBEV disease.

### 6.2. POWV

Human autopsy reports have also shed light on the role of the T cell compartment during POWV disease. In one fatal case of POWV-Spooner infection, the authors noted that a majority of the T cells localized to the meninges were CD4+ T helper cells. In contrast, most T cells localized in the brain parenchyma were CD8+ T cells [136]. These CD8+ T cells were found in close juxtaposition with the neurons remaining at the autopsy, as the patient had suffered severe neuronal loss in the brain and spinal cord, which is common in POWV disease [136,137]. It has also been shown in laboratory mice [74] and outbred strains of mice [74,75] that POWV can establish a lytic infection of neurons in the brain and spinal cord. This recapitulates aspects of the POWV disease in non-human primates [71] and humans [136,138]. Overall, though, the preponderance of evidence in both small animal models and human case studies supports the need to better understand the role of T cells in neuroinvasive POWV disease.

## 7. Long-Term Neurological Sequelae, Persistence and Resolution

### 7.1. TBEV

For some 20–50% of the clinical cases of TBEV, some form of long-term neurological involvement has been documented [33,139], though this is likely to vary depending upon the infecting TBEV subtype. In general, TBEV-Eu is considered to be milder than TBEV-Sib or TBEV-FE subtypes [33], and the risk of incomplete recovery appears to be greater in patients presenting with encephalitis, a more severe form of TBEV [33,140]. Furthermore, the TBEV-Sib subtype appears to be associated with more chronic forms of TBEV, wherein the incubation period can be as long as a year from the time of the tick bite [29]. Some of the earliest characterized TBEV-Sib isolates were generated from chronically infected patients [141]. Though TBEV infection can be persistently detected in some cases, whether the persistence of the infectious virus is the cause of long-term sequelae remains unknown. It is worth noting that the TBEV-Sib subtype was also one wherein previously identified CD8+ T cell epitopes for TBEV-Eu were not conserved [44]. While it is tempting to speculate that this may lead to reduced viral control, more studies of TBEV-specific T cell responses will be needed to address how subtype-specific immunity may impact disease progression.

In follow-up studies of recovered TBEV patients, the most reported symptoms 2–15 years following infection were cognitive or neurological. Reports of lower performance relative to controls in the areas of short- and long-term memory, attentiveness, ability to focus on a task, fatigue, coordination, and in some cases, fine motor skills, were documented [142]. Headaches and sleeping difficulties have also been reported [139,142], as well as neurological symptoms such as limb paresis [33,139].

Muscle atrophy can also result following TBEV infection, with some patients reporting persistent issues 35+ years following infection [143]. In many cases, chronic disease following TBEV infection or relapse can clinically resemble Amyotrophic Lateral Sclerosis (ALS) [143]. There have also been documented cases of fatal progressive TBEV disease, although these remain relatively rare [140]. There may also be an increased risk in TBEV-infected children of incomplete recovery, even in cases where the disease is considered mild [144].

Overall, the causes of neurological sequelae following TBEV infection remain elusive. It is likely that factors including, but not limited to: the infecting subtype, incubation period, degree of neurological involvement, and time to resolution of the disease all impact the likelihood of developing long-term neurological symptoms. As the T cell compartment has been implicated in both viral clearance and pathology in the host, the possibility that patients experiencing long-term neurological sequelae present with some dysfunction of the T cell compartment is plausible and should be considered in future follow-up studies.

Options for addressing the TBEV resolution in animal models remain limited. In early studies of TBEV infection in *Rhesus macaques*, the authors were able to culture an infectious virus isolated from the brains, spinal cord, and peripheral tissues up to 783 days post-infection [54,55,56,57]. These authors also noted that the resolution of motor disorders was not always accompanied by virus elimination, as virus isolates were still recoverable in the case of persistently infected monkeys. The infection of these macaques also generated a detectable T cell response as assayed by splenic migration inhibition in the presence of a TBEV antigen. Further studies of the T cell compartment using non-human primate models of TBEV resolution have yet to be undertaken.

There is limited documented evidence that TBEV can establish persistent infection of primary human cells in vitro. TBEV has been reported to establish persistent infections of primary human brain microvascular endothelial cells (hBMECs) [97]. However, it is worth noting that it is challenging to detect the antigen in brain vasculature in a human autopsy, as immunohistochemical analysis of post-mortem brain sections collected from human TBEV patients has revealed no detection of the TBEV antigen in endothelial cells [129].

Serological evidence suggests that human abortive TBEV cases are possible, though rare [145,146]. For many patients, neuroinvasive TBEV illness resolves after a biphasic illness. Regarding the T cell compartment, this typically occurs in a window during which the T cell response is active. In studies of human infection, Blom et al., found that CD8+ effector T cells transitioned from an effector T (T_eff_) cell phenotype (Eomes^+^Ki67^+^T-bet^+^) to that of a T central memory phenotype (T_CM_) (Eomes-Ki67-T-bet^+^) in the period between 21 and 90 days following hospitalization, consistent with the resolution of infection and contraction of the CD8+ T cell response [52]. Concomitant with this decrease, TBEV-specific T cells were not detected in the PBMCs of these patients at 90 days post-hospitalization. This suggests that following resolution, the TBEV-specific T cell compartment contracts to below the limit of detection for these assays. Studies examining the frequency and function of T_RM_ cells following TBEV infection would further improve our understanding of the T cell compartment during resolution.

Overall, studies of the TBEV resolution, while limited in animal models, indicate that the resolution of biphasic illness with neurological involvement may be partly attributable to T cell responses, as these responses are primed and active during the resolution of disease and eventually progress to more of a memory phenotype.

### 7.2. POWV

In human POWV disease cases, long-term neurological sequelae have been documented for ~50% of patients [42]. Symptoms can include dizziness, headaches, fatigue, coordination, memory issues, muscle weakness, and paralysis [147,148,149,150]. The reported 50% is likely an overestimation, as serological surveys have indicated that POWV infection, much like TBEV, can be resolved asymptomatically for a proportion of hosts [41,151,152]. Much like TBEV, the cause of long-term neurological involvement for POWV remains unclear. In addition, like the related neurotropic flavivirus, the potential for POWV-specific T cells to control a viral infection at the expense of host CNS tissue highlights the need to examine the T cell compartment in follow-up studies of patients surviving neuroinvasive POWV disease.

The high mortality of POWV-challenged laboratory mice presents a barrier to studying long-term neurological sequelae. As such, there is little evidence to suggest that POWV infection can persist in mice or drive long-term neurological sequelae. In laboratory mice surviving a POWV challenge, there are minimal data suggesting cognitive or behavioral changes. In reported natural hosts such as *P. leucopus* mice, lesions and signs of neuronal injury are not detected at more than 30 DPI [74].

Much of the work regarding persistent POWV infection has been carried out in vitro rather than using animal models. POWV has been shown to establish persistent infection in primary human brain microvascular endothelial cells (hBMECs) and pericytes [72], though a POWV antigen has not been detected in these sites during autopsy [136]. Characterization of the POWV-specific T cell response during resolution of the POWV infection has been extremely limited in small animal models and non-existent in human samples. Most of our understanding of a failure to resolve the POWV disease comes from case studies of patients with progressive POWV infection. Notably, two recent progressive fatal cases of POWV infection in two individuals with chronic lymphocytic leukemia (CLL) have been described [153]. Although it is well-established that CLL patients are more susceptible to viral infection and T cell dysfunction [154], the exact nature of those T cell defects and the extent to which this could impact flavivirus encephalitis outcomes remain unknown. Furthermore, both patients presented with high levels of lymphocyte infiltration into the CNS, suggesting that perhaps the inappropriate activation of B and T cells was occurring. Interestingly, intravenous immunoglobulin (IVIG) administration had little appreciable impact on one patient’s status [153], implying that at this point in the patient’s disease progression, IVIG provided little benefit.

Overall, the limited evidence to suggest that POWV disease resolution is at least in part supported by a functional T cell compartment comes indirectly from case reports, wherein immune-compromised patients fail to resolve the infection and the illness progressively worsens despite the administration of IVIG. Many more tools to examine the POWV-specific T cell response in humans and small animal models will be needed before studies of the resolution of POWV infection can proceed.

## 8. Epitope Specific Response

There has been an effort in recent years to characterize the T cell responses to TBFVs, as T cells are known to be activated in human TBFV infection [129]. In general, data regarding the TBEV T cell responses have mostly been obtained from human cohort studies. For TBEV, much of the T cell epitope mapping studies have been carried out in humans. Figure 4 shows a summary of known TBEV epitopes in humans or murine models. In the context of CD4+ helper T cell responses, one study of TBEV vaccinees showed polyfunctional T cell responses to several peptides within the E protein [155]. Although these assays were conducted with peptide pools, vaccinees had detectable T cell responses—in the form of the production of one or more effector cytokines by flow cytometric analysis—to eight out of eleven E peptide pools. This finding implies that several immunogenic peptides could be contained in the E protein, which merits further consideration, especially as these epitopes are likely to be displayed in both infection and immunization.

In one of the first TBEV epitope mapping efforts, Blom and coauthors utilized eleven patient samples (PBMCs) from clinically diagnosed and hospitalized TBEV patients to identify an HLA-A2-restricted TBEV epitope within the NS3 protein [52]. In follow-up studies, Lampen and coauthors utilized PBMCs from five HLA-A2 donors or five HLA-B7 donors, all of which were hospitalized TBEV patients. In addition to re-confirming the previously identified HLA-A2 epitope, the authors identified six additional epitopes, all located within the non-structural proteins [44]. Notably, while these epitopes were all identified using the TBEV-Neudoerfl strain of the European subtype, five out of the seven identified epitopes were highly conserved among the Siberian and Asian TBEV subtypes. This finding was significant, as it highlighted the highly conserved nature of the regions containing T cell epitopes, which is particularly poignant when developing vaccinations that are protective across the TBEV subtypes. Furthermore, this finding also signals the potential for cross-reactive T cell responses, both among TBEV, but also among other members of the TBE serocomplex (i.e., POWV, Langat virus). The question of whether cross-reactive T cell responses can be detected among members of the TBE serocomplex merits further study, as the potential for cross-reactive T cells to potentiate protection or immune pathology has been recognized among mosquito-borne flaviviruses, which are closely antigenically related [156,157].

Our group has previously identified and published murine H2-b restricted T cell epitopes in the structural POWV proteins [77], however, more work will be needed to further identify epitopes located within the non-structural proteins. As many epitopes have been identified within the European subtypes of TBEV NS proteins [44]—and POWV shares roughly 75–76% amino acid identity with the closest known TBEV relatives (Far Eastern subtype, Oshima strain)—further investigation of NS epitopes within POWV lineages merit further consideration. Moreover, studies of the POWV-specific T cell response in humans remain an elusive aspect of our understanding of POWV adaptive immunity. Figure 4 and Table 1 show a summary of known POWV epitopes in humans or murine models. Note that there have been no described human or non-human primate studies of POWV-specific T cell responses.

## 9. Vaccinations and Therapeutics

### 9.1. TBEV

There are currently six vaccines against TBEV approved for use in humans in Europe, Russia, and China and one FDA-approved TBEV vaccine (TICOVAC) (Table 2). In general, all are inactivated, whole virus vaccines. Except for FSME-IMMUN/TICOVAC (TBEV-Eu, Pfizer), all are based on TBEV-FE subtypes. Although historically, viruses to be used as vaccine antigens have been grown in primary chicken embryonic cells, recent attempts to produce the virus in Vero cells appear to be successful [158]. Vaccination typically consists of a primary vaccination containing 2–3 doses, followed by boosting every 3–5 years for individuals considered high-risk [30]. Although rare, failures of TBEV vaccination have been documented to be ~5% and are most common among individuals who are elderly or immune-compromised [159,160]. For this reason, it is suggested that elderly populations receive additional booster doses prior to travel to TBEV endemic areas [160,161]. In addition, it has been well-documented that aging is associated with decreased T cell responses to flavivirus vaccines [162] and changes in BBB permeability [78]. However, the extent to which these factors contribute to TBEV vaccine failure is unknown.

T cell responses are seldom evaluated in response to TBEV vaccination and the mean neutralizing titer is the most commonly evaluated endpoint following vaccination. Even in pre-clinical vaccine efficacy models, research on the T cell responses is scant. Therefore, this section will cover available data on T cell responses to TBEV vaccines, primarily FSME-IMMUN, for which the most data are available.

One study of CD4+ helper T cell responses among FSME-IMMUN vaccinees vs. infected donors noted that the responses following infection tended to be more polyfunctional than those following vaccination [155]. In addition, vaccination tended to result in monofunctional CD4+ T cell responses. Notably, however, the breadth of these responses was retained, with vaccinees and infected donors responding to a similar number of TBEV-Eu envelope peptide pools. Furthermore, despite the importance of CD4+ T cell help in driving the antibody responses, there has been relatively little work characterizing T-follicular helper or T_H_1 responses with respect to TBEV vaccination [155].

Studies of FSME-IMMUN vaccination have also identified differences in donor T cell responses on the basis of biological sex and obesity [164]. PBMCs derived from obese populations and stimulated with TBEV antigen were presented with higher IFN-γ and IL-2 levels. The study also found an overall reduction in CD3+ and CD8+ T cells and lower antibody titers in populations that are obese. This is in accordance with published findings regarding flavivirus vaccination [165,166], as well as work from our group demonstrating that metabolic dysfunction and biological sex can significantly impact WNV vaccination outcomes and T cell responses, especially in the context of memory T cell formation [167].

One study of FSME-IMMUN non-responders demonstrated that—upon stimulation of PBMCs collected after boost—IFN-γ production remained low in the non-responders [164]. The same trend was observed with IL-2, suggesting that non-responders received no benefit from boosting compared to responders. Notably, this phenomenon appeared to be specific to FSME-IMMUN, as the non-responding group did see a benefit from Influenza vaccination that was administered at the same time. Curiously, however, non-responders had a lower proportion of naïve T cells, but higher T memory and T effector memory percentages, both prior to and after boosting. There was a slight but significant trend towards higher Treg responses as measured by FoxP3 expression and IL-10 production in non-responders relative to responders. In addition, the CD8+ T cells in non-responders were described as more highly differentiated as assessed by CD27 and CD28 expression compared to responders.

Another study that compared T cell responses following the subcutaneous or intramuscular administration of FSME-IMMUN found no differences in IL-2, IFN-γ, or IL-10 production following the stimulation of PBMCs with the TBEV antigen, suggesting that the route of administration did not significantly impact the development of the T cell response [168].

Although functional readouts of T cell responses are seldom used in evaluating TBEV vaccine efficacy, these studies indicate a role for cytokine production within the T cell compartment as being important in responding vs. non-responding groups, and support the need for a more thorough characterization of the T cell response to TBEV vaccination. This is particularly important as documented CD4+ and CD8+ T cell responses to the E protein have been observed in humans [44,155] and are contained in the TBEV vaccine antigen, highlighting the need to understand the implication of pre-existing E-specific TBEV immunity in vaccinated populations.

### 9.2. POWV

There exist efficacious vaccines for members of the tick-borne encephalitis serogroup, of which POWV is a member. However, these vaccines are not thought to provide protection against POWV, though the reasons for this have yet to be elucidated [169,170].

There are no licensed POWV vaccines approved for use in humans, nor are there any vaccine candidates in clinical trials at the time of this publication. Several successful vaccination strategies, however, have been tested in small animal models. These are summarized in Table 3. 

In general, nucleic acid-based (mRNA: [70] DNA: [70]), virus-like particle-based [77,172], and nanoparticle-based [69] vaccination strategies have been shown to be efficacious in murine models. Monoclonal antibodies elicited by vaccination have been demonstrated to be important for protection against POWV infection [63,70,173]. In particular, antibodies directed against the POWV envelope domain III have been shown to be protective [69]. Although recent studies have described a role of T cells as being important for the viral control upon the recurrent challenge with POWV [77], only two studies have evaluated T cell responses following vaccination [70,77]. While vaccination did result in functional CD8+ T cell responses as measured by IFN-γ and TNF-α production [77], more thorough phenotypic and functional studies are needed, as well as insights into the kinetics of the response. At the time of this publication, we are aware of no publications addressing the role of either T-follicular helper or T_H_1 responses with respect to POWV infection or vaccination.

## 10. Concluding Remarks

Though TBFVs emerged thousands of years ago, we have only recently begun to appreciate the complex interplay between the virulence, vector biology, host immune status, and prior adaptive immune exposure that dictates the disease outcome. This is particularly true in the case of the role of the T cell response—where the need to immediately process whole blood samples followed by lengthy stimulations for functional readouts has drastically gated the progress in this area. Nonetheless, significant strides have been made in understanding how T cells function in TBFV infection, always toeing the line between protection and pathogenesis. In general, it seems that a potent neutralizing antibody response—in conjunction with peripheral T cell responses—are integral to restricting the entry of TBFV into the CNS and mitigating immune pathology. As more discoveries in the area of TBFV immunology emerge, an eye toward T cell responses will be critical to ensuring the safety and efficacy of vaccines and evaluating the potential intervention strategies to improve TBFV disease outcomes for patients.

## Figures and Tables

**Figure 1 viruses-15-00958-f001:**
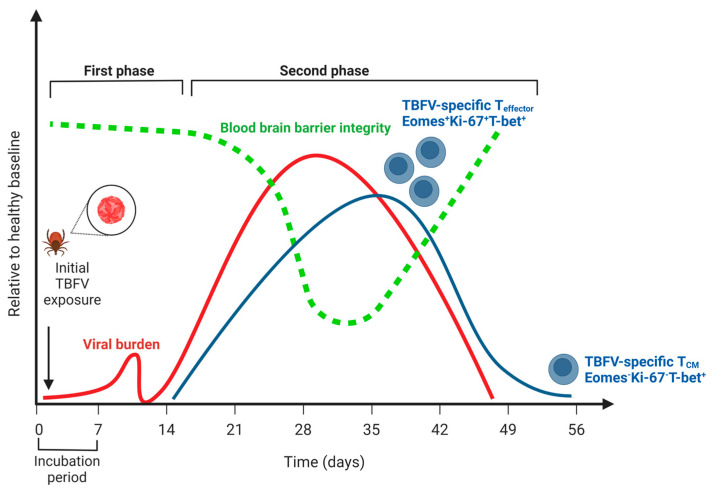
The course of human disease during tick-borne flavivirus (TBFV) encephalitis relative to a healthy baseline. Viral burden is indicated by the red line; note that no distinction is made between peripheral viremia (i.e., during the first phase of disease) and viral replication in the central nervous system (i.e., during the second phase of disease). Blood-brain barrier integrity is indicated by the green line, and notably does not decline until after viremia in the CNS has begun. The T cell response is indicated by the blue line, and the distinction between effector T cells (Eomes^+^Ki-67^+^T-bet^+^) and central memory (Eomes^-^Ki-67^-^T-bet^+^) is highlighted (see Blom, 2018, 30319632). Note that a low frequency of memory cells persists following CNS viral replication and symptoms. Created with BioRender.com (accessed on 31 January 2023).

**Figure 2 viruses-15-00958-f002:**
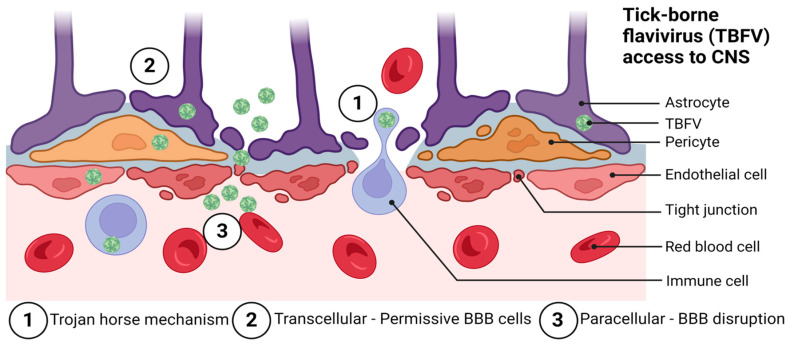
Possible routes of tick-borne flavivirus (TBFV) neuroinvasion and access to the central nervous system through the blood–brain barrier (BBB). These include: (1) the Trojan horse mechanism, (2) transcellular entry, and (3) paracellular entry. Not pictured are routes of entry involving direct infection of the olfactory bulb and retrograde axonal transport. Adapted from: Chen, Z.; Li, G. Immune response and blood-brain barrier dysfunction during viral neuroinvasion. Innate Immun. 2021, 27, 109–117. Created with BioRender.com, accessed on 31 January 2023.

**Figure 3 viruses-15-00958-f003:**
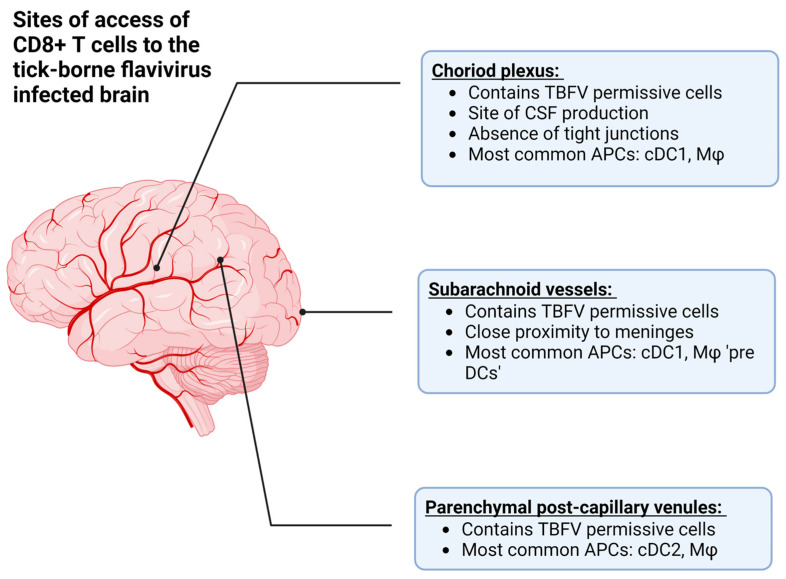
The approximate anatomical locations where T cells can access the central nervous system (CNS) in order to combat tick-borne flavivirus (TBFV) infection. Additionally, this figure describes the antigen presenting cells (APCs) located at each site, such as conventional dendritic cells (type 1 or 2, cDC1 or cDC2), or macrophages (Mφ). Among these are the choroid plexus, which is the site of cerebrospinal fluid (CSF) production, the subarachnoid vessels, and parenchymal post-capillary venules. Created with BioRender.com, accessed on 31 January 2023.

**Figure 4 viruses-15-00958-f004:**
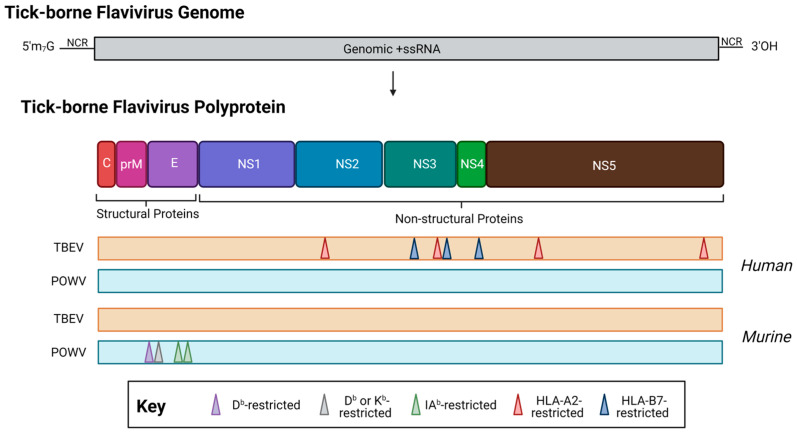
The location of T cell epitopes identified for tick-borne encephalitis virus (orange, TBEV, European subtype, Neudörfl strain) and Powassan virus (teal, POWV, lineage I, POWV-LB) in human (top) or murine (bottom) infection. When possible, MHC restriction is indicated. Note: drawing is not to scale, and regions are approximate. For a detailed location of epitopes relative to the polyprotein, see Table 1. Created with BioRender.com, accessed on 31 January 2023.

**Table 1 viruses-15-00958-t001:** Table 1 shows the location of known tick-borne encephalitis virus (TBEV) or Powassan virus (POWV) T cell epitopes in murine or human infection.

Virus	Organism	Epitope	Amino acid Sequence	MHC Restriction	Virus Reported	Refs.	Notes
TBEV	Human	NS3_1984-1992_	ILLDNITTL	HLA-A2	TBE EK-328 (Siberian subtype)	[44,52]	
	Human	NS2a_1207-1215_	MLLQAVFEL	HLA-A2	TBEV-Neudoerfl	[44]	Not conserved among Siberian and Asian strains
	Human	NS5_2831-2839_	SLINGVVKL	HLA-A2	TBEV-Neudoerfl	[44]	
	Human	NS5_3374-3382 _	NIWGAVEKV	HLA-A2	TBEV-Neudoerfl	[44]	
	Human	NS3_2084-2092_	RPVWKDARM	HLA-B7	TBEV-Neudoerfl	[44]	Not conserved among Siberian and Asian strains
	Human	NS3_1734-1742_	RVRFHSPAV	HLA-B7	TBEV-Neudoerfl	[44]	
	Human	NS4b_2496-2504_	LPLGHRLWL	HLA-B7	TBEV-Neudoerfl	[44]	
POWV	Murine	E_282-291_	THLENRDFV	H2-D^b^	POWV-LB	[69]	
	Murine	E_351-361_	RCPTTGPATL	H2-D^b^ or H2-K^b^	POWV-LB	[69]	
	Murine	E_525-535_	EFGPPHAVKM	I-A^b^	POWV-LB	[69]	
	Murine	E_631-641_	HGVPAVNVAM	I-A^b^	POWV-LB	[69]	

**Table 2 viruses-15-00958-t002:** Summary of tick-borne encephalitis vaccines approved for use in the United States, Europe, and Russia. When available, references that evaluate T cell responses are given. Adult doses only are given. Notes: Encepur (GSK) was divested at the end of 2019. FSME-IMMUN was approved to be marketed in the United States under the name TICOVAC in 2021.

Subtype(s)	Vaccine Name	Manufacturer	Vaccine Type	Target Antigen(s)	Schedule	Route/Dose	T Cell Response
Neudoerf (TBEV-Eu)	FSME-IMMUN/ TICOVAC	Pfizer	Inactivated whole virus	Whole virus	3 doses with boosting optional after 3 years	i.m., 2.4 µg	Y- [155]
Sofjin (TBEV-FE)	TBEV-Moscow	Chumakov FSC R&D IBP RAS	Inactivated whole virus	Whole virus	Prime/boost	i.m., 1.0 ± 0.5 μg/mL	N
Sofjin (TBEV-FE)	Evervac (Phase I/II)	Chumakov FSC R&D IBP RAS	Inactivated whole virus	Whole virus	Prime/boost	i.m., 0.75 ± 0.15 μg	N
Sofjin (TBEV-FE)	Tick-E-Vac	Chumakov FSC R&D IBP RAS	Inactivated whole virus	Whole virus	Prime/boost	i.m. 0.45 ± 0.05 μg	Y- [163]
205 (TBEV-FE)	EnceVir	Microgen	Inactivated whole virus	Whole virus	Prime/boost	i.m., 2.0–2.5 μg	N
Sen-Zhang (TBEV-FE)	SenTaiBao	Changchun Institute of Biol. Products	Inactivated whole virus	Whole virus	Prime/boost	Unknown	N

**Table 3 viruses-15-00958-t003:** Powassan virus vaccine strategies evaluated in vitro or in vivo using small animal models. When available, references that evaluate T cell responses are given.

Lineage	Vaccine Name	Vaccine Type	Target Antigen(s)	Strategy	Route/Dose	Protective	Antibody Response	T Cell Response	Refs.
Spooner	POWV_sig_	mRNA-LNP	prM, E	Prime/ boost	i.m., 10 µg	Y	Y	N	[171]
LB and Spooner	POWV-SEV	Synthetic enhanced DNA	prM, E	Prime/ boost	i.m. + electroporation, 25 µg	Y	Y	Y	[170]
LB	POW-VLP	Virus-like particle	prM, E	Prime/ Boost 1/ Boost 2	i.m., 50 µL	Unknown	Y	N	[172]
LB	POWV-VLP	Virus-like particle	prM, E	Prime/ boost	i.m. 2 µg	Y	Y	Y	[69]
Spooner	LS-POWV-EDIII	Protein + nanoparticle	E (DIII)	Prime/ Boost 1/ Boost 2	i.p. 15 μg	Partially	Y	N	[69]

## Data Availability

Not applicable.

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
