# Peer review of "T Cells in Tick-Borne Flavivirus Encephalitis: A Review of Current Paradigms in Protection and Disease Pathology"

_viruses, 2023, doi:10.3390/v15040958_

Round 1

Reviewer 1 Report

In the manuscript by Stone and Pinto, authors review the role of T-cells in multiple aspects of TBEV and POWV pathogenesis including control of acute infection, mediating immunopathology and their role in vaccine mediated protection. The manuscript is well written and organized in a logical fashion. Authors also provide sufficient background to bring readers up to speed on TBEV/POWV disease. I have no major comments and just a few minor comments for potential additional discussion/clarification. 

Minor comments:

The structure and replication section could be omitted for brevity.

For figure 1, although authors mention it in the legend, instead of labeling the red line viremia, "viral load" or "viral burden" would be more appropriate. Or altering the red line in the second phase to a dashed line to make it more obvious that it refers to both viremia and viral loads in the CNS. It would also be worth including the antibody response on this diagram to highlight that CNS infection is not strictly due to a failure of the host to mount an initial antibody response.

Section 1 could benefit from a geographical map for the viruses and their lineages to help orient readers to the descriptions in the text.

For TBEV and POWV is there anything known on the role of T-cells in support of T-dependent antibody responses? Are antibody responses to these infections in naïve animals or patients dependent on T-help? What about antibody responses to vaccines? If not much is known it would be worth a sentence or two stating as much. Similarly, is there anything known about the role of regulatory T-cells?

In the mouse models has there been attempts to investigate mice deficient or depleted in T-cell effector functions and the impact on TBEV/POWV pathogenesis? For example infection of IFNy or Perforin KO mice? Or altering the timing of depletions to coincide with the early or late phase of disease?

Although authors discuss findings on the role of T-cells in vaccine mediated protection in human subjects, particularly for TBEV has there been any study of the role of T-cells in the pre-clinical vaccine efficacy models?

Author Response

Response to Reviewer 1

We thank the reviewer for their positive assessment of the manuscript. Below we have addressed each of the minor comments suggested by the reviewer in green. Additionally we have attached a copy of the responses.

In the manuscript by Stone and Pinto, authors review the role of T-cells in multiple aspects of TBEV and POWV pathogenesis including control of acute infection, mediating immunopathology and their role in vaccine mediated protection. The manuscript is well written and organized in a logical fashion. Authors also provide sufficient background to bring readers up to speed on TBEV/POWV disease. I have no major comments and just a few minor comments for potential additional discussion/clarification.

We thank this reviewer for this favorable assessment of our manuscript and its relevance to the readership.

Minor

  1. The structure and replication section could be omitted for brevity. We agree with the reviewer that the section on the structure is not strictly related to T cell responses against tick-borne flaviviruses; however several of the other reviewers noted that this section was helpful in flavivirus orientation and asked that we provide more detail. To best accommodate all reviewer requests we have tried to include only the structural detail necessary to understand the T cell responses generated to the tick-borne flaviviruses.
  2. For figure 1, although authors mention it in the legend, instead of labeling the red line viremia, "viral load" or "viral burden" would be more appropriate. Or altering the red line in the second phase to a dashed line to make it more obvious that it refers to both viremia and viral loads in the CNS. It would also be worth including the antibody response on this diagram to highlight that CNS infection is not strictly due to a failure of the host to mount an initial antibody response. Thank you for catching our mistake, at your and that of reviewer #3, we have updated the axes and figure captions. Additionally, we have updated figure one to read ‘viral burden'.
  3. Section 1 could benefit from a geographical map for the viruses and their lineages to help orient readers to the descriptions in the text. We agree that the geographic distribution of TBFVs is pertinent to this review, and as such, have included references to several recent reviews on this topic: TBEV: PMID: 33818450, POWV: PMID: 30541872
  4. For TBEV and POWV is there anything known on the role of T-cells in support of T-dependent antibody responses? Are antibody responses to these infections in naïve animals or patients dependent on T-help? What about antibody responses to vaccines? If not much is known it would be worth a sentence or two stating as much. Similarly, is there anything known about the role of regulatory T-cells? We thank the reviewer for raising several poignant questions. We have added statement on the role on Tregs, reference 58 and included some other the literature that is known about the Th1 response to TBFV, reference 146.  There has been relatively little work characterizing the contribution of CD4+ T cell help with respect to TBEV vaccination, and none that we are currently aware of addressing POWV vaccination. Based on the reviewer's suggestion, we have added statements reflecting this in the vaccination sections.
  5. In the mouse models has there been attempts to investigate mice deficient or depleted in T-cell effector functions and the impact on TBEV/POWV pathogenesis? For example infection of IFNy or Perforin KO mice? Or altering the timing of depletions to coincide with the early or late phase of disease? We thank the reviewer for raising this important question. The authors are currently in the process of drafting a manuscript addressing these questions in mouse models of POWV, and are unaware of other studies of TBEV mouse models addressing this question.
  6. Although authors discuss findings on the role of T-cells in vaccine mediated protection in human subjects, particularly for TBEV has there been any study of the role of T-cells in the pre-clinical vaccine efficacy models? Our finding has been that T cell research, even pre-clinical vaccine efficacy models, is limited, and we have added a statement reflecting this in the vaccination section.

Reviewer 2 Report

The title of this review is:"The role of T cells in tick-borne flavivirus encephalitis", but I consider that the implications in the physiopathogenesis and development of vaccines should be included in it, since they are treated in the document. For the rest it seems to me an excellent document that allows the reader an update on this important topic.

Author Response

Response to Reviewer 2

We thank the reviewer for their positive assessment of the manuscript. Below we have addressed the comments suggested by the reviewer in green.

  1. The title of this review is:"The role of T cells in tick-borne flavivirus encephalitis", but I consider that the implications in the physiopathogenesis and development of vaccines should be included in it, since they are treated in the document. We thank the reviewer for this suggestion and have updated the title to reflect the consideration of pathogenesis and vaccine development to now read “T cells in tick-borne flavivirus encephalitis; a review of current paradigms in protection and disease pathology “.
  2. For the rest it seems to me an excellent document that allows the reader an update on this important topic. We thank this reviewer for this favorable assessment of our manuscript and its relevance to the readership.

Reviewer 3 Report

Manuscript untitled „The role of T cells in tick-borne flavivirus encephalitis” is a nice review about immunological response connected with T cells during TBEV infection.

Article is divided into 10 parts and well organized. Authors starts with chapter presentig flaviviruses. Main part is presentation of tick-borne encephalitis viris (TBEV) and Powassan virus (POWV).

Authors described flaviviruses replication, structure and type of clinical presentation. Even long term incubation period was mentioned connected with FE- subtype of TBEV.

In next chapters ealy T cell response againt viruses are described as well as epitope specific response. In details immunological pathways and reaction are presented.

Large part is about actual knowledge and possibilities of vaccination against TBEV and POWV and its mechanisms of working.

Authors concluded that a potent neutralizing antibody response—in conjunction with peripheral T cell responses—are integral to restricting entry of TBFV into the CNS and mitigating immune pathology.

To sum up, I give a positive opinion about manuscript untitled „The role of T cells in tick-borne flavivirus encephalitis”.

Author Response

Response to Reviewer 3

Manuscript untitled „The role of T cells in tick-borne flavivirus encephalitis” is a nice review about immunological response connected with T cells during TBEV infection.   We thank this reviewer for this favorable assessment of our manuscript and its relevance to the readership.

Article is divided into 10 parts and well organized. Authors starts with chapter presenting flaviviruses. Main part is presentation of tick-borne encephalitis virus (TBEV) and Powassan virus (POWV).  We thank this reviewer for this favorable assessment of our manuscript and its relevance to the readership.

Authors described flaviviruses replication, structure and type of clinical presentation. Even long term incubation period was mentioned connected with FE- subtype of TBEV.  We thank this reviewer for this favorable assessment of our manuscript. 

In next chapters ealy T cell response against viruses are described, as well as epitope specific response. In the details, immunological pathways and reactions are presented. We thank this reviewer for this favorable assessment of our manuscript. 

Large part is about actual knowledge and possibilities of vaccination against TBEV and POWV and its mechanisms of working.  We thank this reviewer for this favorable assessment of our manuscript and its relevance to the readership.

Authors concluded that a potent neutralizing antibody response—in conjunction with peripheral T cell responses—are integral to restricting entry of TBFV into the CNS and mitigating immune pathology. We thank this reviewer for this favorable assessment of our manuscript and its relevance to the readership.

To sum up, I give a positive opinion about the manuscript untitled „The role of T cells in tick-borne flavivirus encephalitis”.     We thank this reviewer for this favorable assessment of our manuscript and its relevance to the readership.

Reviewer 4 Report

The manuscript is aimed at comprehensive analysis of involvement of both innate and adaptive T-cell immunity in pathogenesis of tick-borne flavivirus infections. The topic is relevant and reasonable. However, the paper needs some changes in the way it is written. Before published, some problems should be solved.

Abstract

is not focused on the goal. The main part of the Abstract is devoted to "Introduction" without a description of the aim, main results and final conclusions.

The whole structure and organization of the manuscript is not always clear. Unfortunately, pathogen-associated molecular patterns (PAMP) typical for the tick-borne flaviviruses and the pattern recognition receptors (PRR) that are necessary for innate immunity as the first line defence are not even mentioned. Specific receptors are also important for penetration of the flaviviruses into the central nervous system but details are missing.

Epidemiology in the section 1 is not exact. Thus, there are 6 continents but not 7.

Section 1.1.

The statement "TBEV is transmitted to humans from the bite of Ixodes ricinus or Ixodes persulcatus ticks and is endemic in Europe and Asia" is wrong. Indeed, natural TBEV infection had been revealed for 16 species of ixodid ticks. In Central  Europe TBEV was revealed in 8 species of “hard” ticks having a dorsal shield: Ixodes persulcatus Schulze, Ixodes ricinus L., Ixodes hexagonus, Ixodes arboricola, Haemaphysalis punctata, Haemaphysalis concinna Koch, Dermacentor marginatus Sulz. and Dermacentor reticulatus. In Asia ixodid ticks in the TBE endemic regions included I. persulcatus Sch., I. ricinus L., Ixodes pavlovskyi Pom., D. reticulatus Fabr., D. marginatus Sulz., Dermacentor silvarum Ol., Dermacentor nuttalli Ol., H.concinna Koch.

The statement "several woodland mammals have been suggested as natural reservoirs for TBEV" is also incorrect. The ixodid ticks parasitize more than 100 different species of mammals, birds, reptiles and amphibians thus providing the TBEV transmission and the vertebrate reservoir hosts involvement into epizootic process.  For most vertebrate natural hosts, TBEV is apathogenic without any infection manifectations but induce the virus-specific antibodies.

The reference 13 is about TBEV vertical transmission among vertebrate hosts but not "within tick populations" as described in the same section 1.1.

"Co-feeding" is nonviremic transmission (NVT) via co-feeding tick saliva, isn't it?

TBEV does not seem to be "endemic in a large geographical area" but rather in natural foci.

Epidemiology is close to vaccination rates. Therefore,  all available vaccines should be described at the beginning but not at the end of the manuscript.

Immediately after discovery of TBEV in 1930s, a first inactivated vaccine was developed. All seven available vaccines against the tick-borne encephalitis produced in Russia (2), Austria (2), Germany (2) and China (1) proved their safety and efficacy in humans. Currently, Russian vaccine strains of the Far Eastern genetic subtype include strain Sofjin isolated from a patient’s brain in 1937 and strain 205 isolated from the Ixodes persulcatus Schulze tick in Khabarovsk region in 1973. European vaccines are based on strain Neudorfl isolated from an Ixodes ricinus tick in 1971 and strain К23 also originated from I. ricinus in 1975. In China strain Senzhang of the Far Eastern subtype isolated from a patient’s  brain in 1953 is used for vaccine production by the Changchun Institute of Biological Products [6]. Thus, all currently available vaccines against the tick-borne encephalitis are based on the strains heterologous to the majority of the current TBEV isolates and were isolated more then 50 years ago.

The section 2. Flavivirus structure and replication.

NS1 is located on the surface of infected cells and possess important antigenic determinats too. Not only glycoprotein E.

Biphasic TBE is not typical. Many clinical cases include the only mild fever symptoms.

Figure 1. The term "viremia" is commonly used for the virus in blood. The virenic period is short and unpredictable. It's hardly possible to isolate viable TBEV from animal blood.

"Outbred mice" cannot  serve as natural hosts for TBEV since they are not adapted during long virus-host co-evolution.

The section 7. "Long-term neurological sequelae, persistence and resolution"

There are multiple evidences of the TBEV persistence in wild mammals. in ticks and in patients.

Other typos and other problem points are highlighted in yellow in the attached file.

Author Response

Response to Reviewer 4

The manuscript is aimed at comprehensive analysis of involvement of both innate and adaptive T-cell immunity in pathogenesis of tick-borne flavivirus infections. The topic is relevant and reasonable. However, the paper needs some changes in the way it is written. Before published, some problems should be solved.

We thank this reviewer for this favorable assessment of our manuscript and have worked to incorporate the suggested changes. We feel that these additions have strengthened the manuscript and improved both the structure and relevance for the readership.

Abstract is not focused on the goal. The main part of the Abstract is devoted to "Introduction" without a description of the aim, main results and final conclusions.     The whole structure and organization of the manuscript is not always clear. We thank the reviewer for their thoughtful review and have revised the abstract to highlight goals and findings. As three out of four reviewers gave positive feedback regarding the organization of the manuscript, we have made revisions to clarify sections and support the scientific assessment of all four reviewers. However, we have, throughout the manuscript made targeted changes to specifically address the reviewer's concern regarding the clarity of the organization.

Unfortunately, pathogen-associated molecular patterns (PAMP) typical for the tick-borne flaviviruses and the pattern recognition receptors (PRR) that are necessary for innate immunity as the first line of defense are not even mentioned.      We agree that PAMPs & PRRs are critical early lines of defense against TBFVs. As the review is focused on the T cell responses to the pathogen, we have elected to focus on innate immunity's importance in driving T cell responses. Thus we have added statements in the sections “T cell responses early in TBFV infection” and “The central nervous system (CNS) and immune-privilege”. These statements reflect the importance of PAMPs & PRRs in potentiating and recruiting T cells. Furthermore, we have directed readers to excellent pertinent references on this important topic [PMID: 32196427; PMID: 30710567; PMID: 29933625; PMID: 28167278; PMID: 27543616; PMID: 16188985].

Specific receptors are also important for penetration of the flaviviruses into the central nervous system but details are missing. While the expression of specific receptors permitting entry into the CNS is a pertinent topic, literature implicating individual receptors as necessary and sufficient for TBFV infection of the CNS, to our knowledge, is scant. We have added a statement clarifying this to the section “Trafficking to and across the blood-brain barrier into the CNS” and have added references [PMID: 24381034; PMID: 19961413; PMID: 25319621; PMID: 1456894; PMID: 10393498] to direct readers to existing reviews on this topic.

Epidemiology in the section 1 is not exact. Thus, there are 6 continents but not 7. The statement has been edited for clarity to read “Encephalitic flaviviruses have been isolated on almost every continent”.

The statement "TBEV is transmitted to humans from the bite of Ixodes ricinus or Ixodes persulcatus ticks and is endemic in Europe and Asia" is wrong…. Koch          We have revised this statement to instead read “TBEV is transmitted to humans from the bite of Ixodes ticks of numerous species and is endemic in Europe and Asia” for clarity.

The statement "several woodland mammals have been suggested as natural reservoirs for TBEV" is also incorrect…antibodies     In existing literature, many mammals have been posited as possible natural TBEV reservoirs, and many current reviews on the topic suggest that a single reservoir has not been implicated. We have added references [PMID: 30234026; PMID: 31336624; PMID: 24349041] to direct readers to existing reviews and articles on this topic.

The reference 13 is about TBEV vertical transmission among vertebrate hosts but not "within tick populations" as described in the same section 1.1. The contribution of transovarial transmission for TBEV is generally thought to be minimal relative to other routes of transmission. This statement has been revised for clarity to read: “It should also be noted that vertical transmission within tick populations may be important in perpetuating the TBEV lifecycle (PMID: 12641208; PMID: 20668521), though the contribution is likely small relative to other routes or in mammalian hosts (13).”

"Co-feeding" is nonviremic transmission (NVT) via co-feeding tick saliva, isn't it? We thank the reviewer for highlighting the opportunity to include this terminology and have updated this statement to read: “Co-feeding—or nonviremic transmission (NVT) facilitated by tick saliva —is thought to be particularly…”.

TBEV does not seem to be "endemic in a large geographical area" but rather in natural foci.         We have revised this statement for clarity to read “Although TBEV infection occurs within a large geographic area, TBEV incidence tends to occur in focal hot spots across a given region [20].”

Epidemiology is close to vaccination rates. Therefore,  all available vaccines should be described at the beginning but not at the end of the manuscript.   We respect all of the reviewers' opinions, as three out of four reviewers gave positive feedback regarding the organization of the manuscript, and this includes the discussion of vaccination after protective immune responses. We have therefore elected not to undertake structural revisions.

Immediately after discovery of TBEV…were isolated more than 50 years ago.     The information regarding TBEV vaccines and the corresponding isolates that are the basis of vaccine antigens can be found in Table 1.

NS1 is located on the surface of infected cells and possesses important antigenic determinants too. Not only glycoprotein E. We have updated this flavivirus structure section to reflect the importance of the adaptive immune response to non-structural proteins to read: “The non-structural proteins, in addition to being indispensable for viral replication, are known to be targets for both the T cell PMID: 31022685 and antibody responses PMID: 31839903; PMID: 23523765; PMID: 35891164.

Biphasic TBE is not typical. Many clinical cases include the only mild fever symptoms. The term ‘biphasic’ was used in regard to the symptoms patients typically experience. As such, we have revised the statement for clarity to read: “In humans, acute symptomatic illness due to TBEV is considered to be  biphasic.”

Figure 1. The term "viremia" is commonly used for the virus in blood. The viremic period is short and unpredictable. It's hardly possible to isolate viable TBEV from animal blood.  At the suggestion of this reviewer and reviewer #1, we have updated the axes and figure captions for figure one to instead read ‘viral burden’.

"Outbred mice" cannot serve as natural hosts for TBEV since they are not adapted during long virus-host co-evolution. We thank the reviewer for allowing us to clarify the terminology used. In the referenced study, certain cell types were permissive to TBEV replication. Thus, this statement has been revised for clarity to read: “…followed by dissemination into secondary lymphoid tissues, as observed in outbred mice, which can support  TBEV replication[14, 15]”

There are multiple evidences of the TBEV persistence in wild mammals. in ticks and in patients.   We agree with the reviewer's assertion and have referenced several studies (see [45-48]) documenting TBEV persistence.

  In the instances where the reviewer highlighted sections. We have gone through the manuscript and have checked the formatting. The other reviewers of the manuscript did not note these issues. We are sorry for the formatting distractions and hope that in the updated version, the reviewer does not encounter these issues. We will be happy to address them as they arise and appreciate the diligence of the reviewer.

Round 2

Reviewer 4 Report

General comments.

Main directions of the manuscript are relevant. According to the title one can expect analysis of T-cell epitopes, their multiple alignment in order to reveal homology levels among TBEV isolates of different subtypes and for different tick-borne flavivirus species. General scheme of the manuscript is not common (from innate immunity to innate-like and adaptive immune response including Th1 T-cell immunity). There is not cellular immune response without a damage of infected and surrounding cells. Therefore, it would be interesting to reveal molecular mimicry or other immunomodulation mechanisms caused by tick-borne flaviviruses (for example, TBEV NS5 protein as regulator of immune response).

Specific comments.

Problem points are highlighted in yellow in the attached file.

The mortality rate from "non-malarial vector-borne diseases worldwide" seems to far from the topic. Both morbidity and mortality rates periodically change. Recent numbers and trends are important to compare. 

"TBEV is transmitted to humans from the bite of Ixodes ricinus or Ixodes persulcatus ticks of numerous species and is endemic in Europe and Asia"

How about other hard ticks? North Africa?

Natural TBEV infection had been revealed for 16 species of ixodid ticks. Thus, in Central  Europe TBEV was revealed in 8 species of “hard” ticks having a dorsal shield: Ixodes persulcatus Schulze, Ixodes ricinus L., Ixodes hexagonus, Ixodes arboricola, Haemaphysalis punctata, Haemaphysalis concinna Koch, Dermacentor marginatus Sulz. and Dermacentor reticulatus.

"TBEV exists in five recognized subtypes"

To be exact 3 main subtypes.

TBEV is grouped into seven subtypes according to their phylogenetic relationships. Indeed, virus strains that differ by less than 10% of nucleotides in the polyprotein-coding gene are provisionally proposed to belong to the same subtype. Within the TBEV viral species, three main subtypes are defined by the International Committee on Taxonomy of Viruses: , the European TBEV (TBEV-EU), the Siberian TBEV (TBEV-Sib), and the Far Eastern TBEV (TBEV-FE). Co-circulation of different subtypes was already demonstrated in a hyperendemic area.    Lundkvist, K., Vene, S., Golovljova, I., Mavtchoutko, V., Forsgren, M., Kalnina, V., Plyusnin, A. Characterization of tick-borne
encephalitis virus from Latvia: evidence for co-circulation of three distinct subtypes. J Med Virol. 2001;65(4):730-5. doi:
10.1002/jmv.2097.
 Kubinski, M., Beicht, J., Gerlach, T., Volz, A., Sutter, G., Rimmelzwaan, G.F. Tick-Borne Encephalitis Virus: A Quest for Better
Vaccines against a Virus on the Rise. Vaccines (Basel). 2020; 8(3): 451. doi: 10.3390/vaccines8030451
 Fafangel, M., Cassini, A., Colzani, E., Klavs, I., Grgič Vitek, M., Učakar, V., Muehlen, M. Vudrag, M., Kraigher, A. (2017). Estimating the annual burden of tick-borne encephalitis to inform vaccination policy, Slovenia, 2009 to 2013. Euro Surveill., 22(16), 783
pii=30509. doi: http://dx.doi.org/10.2807/1560-7917.ES.2017.22.16.30509.   "Although TBEV infection occurs within a large geographic area" Tick densities, TBEV infection rate and morbidity differ in the natural focus center and in surrounding regions.   "Ixodes scapularis ticks and is endemic in North America and the Russian Far East" Ixodes scapularis does not inhabit the  Far East of Russia (and China, too).     "envelope protein, the main antigenic determinant" Both structural and non-structural proteins of the TBEV are immunogenic. The glycoprotein E is on the surface of the extracellular virions whereas non-structural glycoprotein NS1 is on the surface of the virus-infected cells.   Fig. 1 is far from its legend. It's hardly possible to find and put them together.  Moreover, "the distinction between effector T cells (Eomes+Ki-67+T-bet+) and central memory (Eomes-Ki-67-T-bet+)" is not shown.   Abbreviations CNS and BBB are not used throughout the text.   Fig. 2 is inserted twice.   "There are currently five vaccines against TBEV approved for use in humans in Europe and Russia and one FDA-approved TBEV vaccine (TICOVAC) (Table 1)" In China strain Senzhang of the Far Eastern subtype isolated from a patient’s  brain in 1953 is used for vaccine production by the Changchun Institute of Biological Products.          

Author Response

Please find attached the second round of reviewer responses for the manuscript entitled: “T cells in tick-borne encephalitis: a review of current paradigms is protection and disease pathology”. We have worked diligently to incorporate the changes suggested. We thank each of the reviewers for taking the time to review our manuscript and feel that the additions made at their suggestion have strengthened the manuscript and improved the relevance of our work for the journal’s readership.  Response to the fourth reviewer's second round of suggestions are below.

Main directions of the manuscript are relevant.        

We thank this reviewer for this favorable assessment of our manuscript and incorporated the suggested changes, which have strengthened the manuscript.

According to the title one can expect analysis of T-cell epitopes, their multiple alignment in order to reveal homology levels among TBEV isolates of different subtypes and for different tick-borne flavivirus species.

 T cell epitopes have been indicated in the subtypes with which they were identified in Figure 4. In their publication, Lampen et. Al. performs this comparison among TBEV subtypes. 

General scheme of the manuscript is not common (from innate immunity to innate-like and adaptive immune response including Th1 T-cell immunity).     

 The article is being submitted for a special issue on "T cell responses to viral infections". We agree that Innate immunity is an important topic. However, as the manuscript and the special issue are devoted to T cells, we did not undertake major structural revisions.

There is not cellular immune response without a damage of infected and surrounding cells. Therefore, it would be interesting to reveal molecular mimicry or other immunomodulation mechanisms caused by tick-borne flaviviruses (for example, TBEV NS5 protein as regulator of immune response).       

We agree that this is topic is of high relevance to the readership, but we feel is beyond the scope of this review. The line of inquiry posed would best be addressed in a research article. 

The mortality rate from "non-malarial vector-borne diseases worldwide" seems to far from the topic.      

We agree that this is a broad categorization and have updated the statement to reflect the burden of flavivirus infection only. Unfortunately, data describing the global public health burden of tick-borne flaviviruses is scarce, and global metrics of tick-borne disease tend to disproportionally represent bacterial diseases (i.e. Lyme disease).

"TBEV is transmitted to humans from the bite of Ixodes ricinus or Ixodes persulcatus ticks of numerous species and is endemic in Europe and Asia" How about other hard ticks? North Africa? Natural TBEV infection had been revealed for 16 species of ixodid ticks. Thus, in Central  Europe TBEV was revealed in 8 species of “hard” ticks having a dorsal shield: Ixodes persulcatus Schulze, Ixodes ricinus L., Ixodes hexagonus, Ixodes arboricola, Haemaphysalis punctata, Haemaphysalis concinna Koch, Dermacentor marginatus Sulz. and Dermacentor reticulatus.

We revised this statement in previous rounds of revision to instead read “TBEV is transmitted to humans from the bite of Ixodes ticks of numerous species and is endemic in Europe and Asia”.

"TBEV exists in five recognized subtypes" To be exact 3 main subtypes. TBEV is grouped into seven subtypes according to their phylogenetic relationships. Indeed, virus strains that differ by less than 10% of nucleotides in the polyprotein-coding gene are provisionally proposed to belong to the same subtype. Within the TBEV viral species, three main subtypes are defined by the International Committee on Taxonomy of Viruses: , the European TBEV (TBEV-EU), the Siberian TBEV (TBEV-Sib), and the Far Eastern TBEV (TBEV-FE). Co-circulation of different subtypes was already demonstrated in a hyperendemic area.          

We have revised this statement to instead read “TBEV exists in 3-7 subtypes, depending upon classification schema” for clarity.

 "Although TBEV infection occurs within a large geographic area" Tick densities, TBEV infection rate and morbidity differ in the natural focus center and in surrounding regions.    

We addressed this statement in previous rounds of revision.

 "Ixodes scapularis ticks and is endemic in North America and the Russian Far East" Ixodes scapularis does not inhabit the  Far East of Russia (and China, too).  

In this sentence, the antecedent (or referent) is Powassan virus, thus the implication is not that Ixodes scapularis ticks inhabit the aforementioned region. We hope that this clarifies the issue.

 "envelope protein, the main antigenic determinant" Both structural and non-structural proteins of the TBEV are immunogenic. The glycoprotein E is on the surface of the extracellular virions whereas non-structural glycoprotein NS1 is on the surface of the virus-infected cells.         

 We addressed this statement in previous rounds of revision. 

Fig. 1 is far from its legend. It's hardly possible to find and put them together.       

 We have attempted to correct this formatting issue and will continue to work with the journal on improving the placement of the figures.

Moreover, "the distinction between effector T cells (Eomes+Ki-67+T-bet+) and central memory (Eomes-Ki-67-T-bet+)" is not shown.

We elected not to separate out the effector and central memory cells, as this would crowd the figure. We have, in many places referred to or directed readers to the primary research article for clarification on this topic.

Abbreviations CNS and BBB are not used throughout the text.       

 We have revised to consistently use these abbreviations

Fig. 2 is inserted twice           

 We have attempted to correct this formatting issue and will continue to work with the journal on improving the placement of the figures.

"There are currently five vaccines against TBEV approved for use in humans in Europe and Russia and one FDA-approved TBEV vaccine (TICOVAC) (Table 1)" In China strain Senzhang of the Far Eastern subtype isolated from a patient’s  brain in 1953 is used for vaccine production by the Changchun Institute of Biological Products.  

 Although we specified that the table was focused primarily on vaccines approved in Europe and Russia, we have updated the table to reflect this information.